# Robust 3D Hand Detection from a Single RGB-D Image in Unconstrained Environments

**DOI:** 10.3390/s20216360

**Published:** 2020-11-07

**Authors:** Chi Xu, Jun Zhou, Wendi Cai, Yunkai Jiang, Yongbo Li, Yi Liu

**Affiliations:** 1School of Automation, China University of Geosciences, Wuhan 430074, China; xuchi@cug.edu.cn (C.X.); caiwendi@cug.edu.cn (W.C.); jiangyunkai@cug.edu.cn (Y.J.); ybli@cug.edu.cn (Y.L.); 2Hubei Key Laboratory of Advanced Control and Intelligent Automation for Complex Systems, Wuhan 430074, China; 3Engineering Research Center of Intelligent Technology for Geo-Exploration, Ministry of Education, Wuhan 430074, China; 4CRRC Zhuzhou Electric Locomotive Co., Ltd., Zhuzhou 412000, China; liuyi_hust@163.com; 5National Innovation Center of Advanced Rail Transit Equipment, Zhuzhou 412000, China

**Keywords:** 3D hand detection, RGB-D sensor, human–computer interaction, unseen lighting condition, adaptive RGB-D fusion

## Abstract

Three-dimensional hand detection from a single RGB-D image is an important technology which supports many useful applications. Practically, it is challenging to robustly detect human hands in unconstrained environments because the RGB-D channels can be affected by many uncontrollable factors, such as light changes. To tackle this problem, we propose a 3D hand detection approach which improves the robustness and accuracy by adaptively fusing the complementary features extracted from the RGB-D channels. Using the fused RGB-D feature, the 2D bounding boxes of hands are detected first, and then the 3D locations along the z-axis are estimated through a cascaded network. Furthermore, we represent a challenging RGB-D hand detection dataset collected in unconstrained environments. Different from previous works which primarily rely on either the RGB or D channel, we adaptively fuse the RGB-D channels for hand detection. Specifically, evaluation results show that the D-channel is crucial for hand detection in unconstrained environments. Our RGB-D fusion-based approach significantly improves the hand detection accuracy from 69.1 to 74.1 comparing to one of the most state-of-the-art RGB-based hand detectors. The existing RGB- or D-based methods are unstable in unseen lighting conditions: in dark conditions, the accuracy of the RGB-based method significantly drops to 48.9, and in back-light conditions, the accuracy of the D-based method dramatically drops to 28.3. Compared with these methods, our RGB-D fusion based approach is much more robust without accuracy degrading, and our detection results are 62.5 and 65.9, respectively, in these two extreme lighting conditions for accuracy.

## 1. Introduction

Hands play an important role in people’s daily activities. Hand detection is a key component in many computer vision applications, such as human–computer interaction [1], hand pose estimation [2,3,4], hand gesture recognition [5,6], activity analysis [5], and so on. Most existing works [7,8,9,10,11,12] focus on 2D hand detection from a single RGB image which lacks 3D information and leads to incompetency for 3D hand detection. However, the real world is of 3D by nature, and the RGB image based methods cannot meet the increasing requirement of 3D human–computer/robot interaction [13]. For example, in a robotic teaching scenario, there would be ambiguities in inferring the target from a single RGB image (see Figure 1a). Therefore, it is necessary to fuse the RGB image with 3D representations extracted from the depth image to enable 3D hand detection. (In this paper, we focus on 3D hand detection in unconstrained environments. For other related technologies such as the estimation of hand joints and action recognition, please refer to our previous work [2,3,14]).

Currently, 3D hand detection from a single RGB-D image in complex unconstrained environments is a challenging task. As can be seen in Figure 2, hand appearances on RGB-channel and D channel can be affected by many factors, such as various light conditions, hand shapes, viewpoints, scales, partial occlusions, and so on.The RGB-channel and the D-channel contain complementary information for hand detection, and the characteristics of these two modalities are very different from each other: The RGB-channel contains rich information of color, shape and textual, but can be significantly affected by light variation; on the contrary, the D channel is much stabler with respect to light variation excepting back-light condition which leads to failure regions (please refer to the D channel in Figure 2d). What’s more, the depth noise increases as the depth value increases.

In order to effectively fuse the RGB-D channels for robust 3D hand detection, the following open issues need to be considered: (i) It is difficult to determine which level(s) of features (i.e., the intermediate results of the CNN stages which are inside the network) are optimal for RGB-D fusion. Normally, only high-level features (output of the last CNN stage of the network) are fused [15,16,17,18,19], while the features on other levels are bypassed. In [20], multi-level features are explored by an exhaustive enumeration scheme, but it is time consuming due to many possible fusion joints (please refer to Figure 3d. (ii) For the D channel, the local representation on hand will be weakened (almost “flattened”) if the depth image is directly normalized using the whole depth range (please refer to Section 3.1), since the whole depth range of unconstrained environment is much bigger than that of constrained environment. It is necessary to explore a new depth representation for hand detection, so that the local representation on hand can be enhanced. (iii) Unconstrained environments are not considered in existing RGB-D hand datasets, which results in insufficiency of evaluating the state-of-the-arts. (The reason primarily contains two aspects: (1) Technology limit. Hand detection is the foundation of many hand related applications such gesture recognition, hand pose estimation and human computer interaction, but the performance of existing hand detector is far from perfect. To reduce the influence caused by unstable hand detection, many datasets for these applications are collected in constrained environment (single person/hand per image, simple background, and so on). If unconstrained environments are considered in these applications, the performance may dramatically degrade. (2) It is hard/costly to collect hands from unconstrained environments which consider various factors. Existing RGB hand detection datasets considering unconstrained environments (e.g., Oxford hand) collect images from web so that the cost can be reduced, but RGB-D images are not commonly available on web.) To better evaluate the RGB-D hand detection in unconstrained environments, more challenging benchmark is required.

In this paper, we propose a RGB-D fusion based approach for 3D hand detection in unconstrained environments. Based on the fused multi-modal features, the 2D hand locations on image plane are detected first, and then the 3D hand locations along the z-axis are estimated by a cascaded 3D location estimator. The main contributions of our work are fourfold:We propose a novel adaptive fusion network (AF-Net) which adaptively fuse multi-level features for 3D hand detection. The core of AF-Net is a cross-modal feature fusion unit named “adaptive fusion unit” (AFU). As can be seen in Figure 3e, AFUs control the connectivity of fusion paths: if the weights of a AFU are set to value 0, the fusion path is blocked, otherwise it is activated. The fusion structures enumerated in Figure 3d can be obtained by adjusting the weights of AFUs. Thus, AF-Net can be regarded as a generalized version of [20]. Instead of exhaustively searching for an optimal joint to fuse the RGB-D branches [20], multi-level features are adaptively fused by AFUs and their weights are optimized in an end-to-end manner. It performs significantly robuster than hand detectors without fusion.We propose a stacked sub-range representation (SSR) for 3D hand detection in unconstrained environments. The whole depth range of the D-channel is evenly divided into a series of smaller stacked sub-ranges, so that the normalized local depth representations within each sub-range can be enhanced (please refer to Section 3.1). The D-channel is transformed to SSR first, and then it is fed into the network for feature extraction, fusion and hand detection. SSR produces much more accurate results than the raw depth representation.We propose a challenging RGB-D hand detection dataset named “CUG Hand”. To the best of our knowledge, it is the first RGB-D hand detection dataset collected in unconstrained environments. Existing RGB-D hand datasets are normally captured indoors, and contain only a single subject (up to 2 hands) per-image, whereas our dataset contains unconstrained environments, the number of subjects varies from 1 to 7 per-image, and the maximum number of hands per-image is up to 8. In order to evaluate the robustness and accuracy of the state-of-the-arts, various challenging factors such as extreme light conditions, hand shape, scale, view point, partial occlusion are considered in this dataset.The proposed 3D hand detection approach is extensively evaluated on CUG Hand dataset, as well as a public RHD hand dataset [21]. Experimental results show that the proposed approach significantly outperforms the stat-of-the-arts in terms of accuracy, and it can robustly detect 3D hand even in extreme light conditions. The proposed approach can have a wide range of hand related applications, such as hand gesture recognition, hand pose estimation, activity analysis, human–computer interaction, and so on.

The CUG Hand dataset and the related code will be publicly released online in the future: https://github.com/cug633/3D-Hand-Detection.

## 2. Related Work

In this section, we briefly review related work regarding 2D hand detection, 3D hand detection, RGB-D fusion based detection methods, and hand detection datasets.

2D hand detection. There exist a long line of literatures that focus on 2D hand detection from RGB images. Traditional approaches [22,23,24,25] mainly use hand-crafted weak features such as HOG [26,27], skin color [7,24,28,29], etc. In recent years, 2D hand detection accuracy has been significantly boosted by deep learning based methods. Le et al. [8] detect hands using a CNN network with multi-scale feature map. Gao et al. [30] combine deep layers with shallow layers for hand detection. In [9,10,11], in-plane hand rotation information is explored to improve the detection precision. In [12], the generalization ability and detection accuracy are enhanced by introducing an auxiliary hand appearance reconstruction task. In this paper, we focus on static hand detection which takes single images as input. It can be applied to video clips, since a video clip can be considered as an image sequence in which each frame can be processed individually. Furthermore, static detection provides initialization for dynamic tracking, and many dynamic tracking methods [31] are conducted based on the results of static detection.

3D hand detection. With the emergence of consumer-level depth sensors [32], depth images have been used for 3D object detection. Kinect [32] estimates 3D locations of hands as well as other body joints, but it requires that upper body parts should be visible in the depth image without much occlusion. In [33,34,35] the target hand is supposed to be the nearest object in the depth image, so that the hand can be easily located by simple image processing. Traditional learning algorithms such as random forest [36] and cascade weak classifiers [37] are also used for hand detection from depth images where hands are the nearest objects to the camera. The methods mentioned above are designed for constrained environment. It is necessary to fuse the depth image with the RGB image for 3D hand detection in unconstrained environments.

RGB-D fusion for detection. Instead of using only RGB or depth images for detection, fusing these complementary modalities can improve the detection performance. In [38,39,40], the RGB-D channels are fused in a two-step scheme: firstly, 2D bounding boxes of objects are located on image plane using RGB image based 2D detector; secondly, the objects’ 3D positions along the z-axis are estimated from the cropped RGB-D image. However, its limit is that important features contained in the D channel are not fused in the first step of 2D bounding box localization.

The RGB-D channels are complementary. In order to effectively fuse the RGB-D channels, recent research has focused on following principal directions:

(1) The first direction is to explore the representation of the RGB-D image. Different types of RGB-D representations are as follows:2D convolutional representations. In [41], the raw depth image (i.e., the D channel) is concatenated with the RGB channels, and then the RGB-D channels are fed into a 2D convolutional network. In [38,42], the depth image is transformed into a 3-channel HHA representation (Height above ground, Horizontal disparity, and Angle with gravity) for semantic segmentation of indoor scenes. In [43], object detection proposals are generated in a top-down bird view which is based on a restrictive assumption that all objects are on the same spatial plane, e.g., cars on road.3D convolutional representations. The RGB-D image can be converted into 3D convolutional representations such as Voxel [44] and TSDF [45]. However, due to the curse of dimensionality, these representations are computationally expensive with large memory footprints. 3D convolutional representations are usually applied in constrained environment within a limited cubic range, e.g., indoor scenes.Point-cloud representations. The depth image can be represented as point-cloud [39] for recognition. The point-cloud representations can be further enhanced by concatenating each point with their corresponding RGB features extracted from CNN [40,46]. These methods follow the two-step scheme mentioned above. As they take the 2D bounding boxes detected from only RGB image as input, the information in the D channel is not fully fused for detection.

The SSR proposed in this paper is a 2D convolutional representation. It is computational efficient and does not rely on any specific assumption, so it can be easily applied in unconstrained environments.

(2) The second direction is to locate which level(s) of feature shall be fused. According to the level of feature, existing methods can be classified into following categories:Early fusion. The RGB-D channels are fused before the images are fed into the CNN [41,47]. The RGB-D channels are directly concatenated, and only low-level features are fused by early fusion.Late fusion. The RGB-D channels are fused at the end of the feature extraction CNN networks [15,16,17,18,19]. The RGB and D branches are trained in parallel and then the features from both modalities are fused at the last stage. High-level features are fused by late fusion, but mid-level features are not fully fused.Intermediate fusion. The RGB-D channels are fused at intermediate stages of the CNN networks [20,48]. In the CNN networks, a single stage or multiple stages are selected at which the RGB and D branches are joined. Mid-level and high-level features are fused by intermediate fusion. However, it is not clear which position is the optimal fusion joint. One solution is to conduct an exhaustive enumeration [20] so that the best position can be found. Another solution [49,50,51] is to progressively fuse the features from one branch to another on multiple corresponding stages. While the later solution is primarily applied in per-pixel classification tasks such as semantic segmentation and salient object detection, and it is seldom used for region proposal based object detection tasks.

Our proposed AF-Net belongs to intermediate fusion. It can be regarded as a generalized version of [20] when the AFU weights corresponding to a specific joint are set to value 1 and that corresponding to other joints are set to value 0. References [49,50,51] also fuse multi-level cross-modal features. However, different from them, the proposed AFU adaptively adjusts the connectivity between the branches, and the unnecessary fusion path can be effectively cutoff. Therefore the AF-Net is robust against “over-fusion”.

(3) The third direction is to investigate the fusion process of multi-modal features. It can be primarily classified into two categories:Basic fusion operator. In [49], the pixel-wise summation operator is used for RGB-D fusion in semantic segmentation application. In [52], basic operators such as concatenation, summation, multiplication, etc. are compared, and it is found that the summation operator works well in the extreme exposure image fusion application.Advanced fusion layer. Instead of directly using basic operators, advanced fusion layers are designed by combining basic operators or sub-networks. In [20], a fusion layer is defined as a combination of a concatenation operator and a 2D convolutional layer. In [53], an fusion layer is proposed by combining a contrast-enhanced sub-network and a pixel-wise multiplication operator for per-pixel salient object detection task. Furthermore, sub-networks such as graph convolutional network [17], gating network [15] and LSTM [19] have been used to construct advanced fusion layers for the high-level features in the late fusion stage. In [54], tree-structured LSTM is used to extract relations between lexical-level features and syntactic features.

Our proposed AFU can be simplified as an ordinary summation operation when the AFU weights are set to a constant value 1. The work most related to our AFU is a splitting unit named cross-stitch unit [55]. They both adaptively learn sharing weights between two related branches. Their differences contain two aspects: (1) the cross-stitch unit focuses on multitask learning problem which splits one network into two sub-branches, while our AFU aims at fusing the information of multi-modal branches into one network; (2) for each pair of features, the cross-stitch unit uniformly adjusts the connectivity of all the channels by scalar weights, whereas the AFU weights are vectors, and the connectivities between each pair of corresponding channels can be adjusted specifically.

Besides, some RGB-D fusion related papers focus on other computer vision tasks such as 3D reconstruction [56], semantic segmentation [49], salient object detection (Salient object detection aims at modeling the attention mechanism of human visual systems, and it is very different from the normal object detection task) [50,51,53], and so on. These works are very different from ours and will not be discussed this paper.

Hand Detection Datasets. Most of the existing hand detection datasets contain only RGB images with 2D labels of hands, such as Oxford hand dataset [7], EgoHand dataset [57], VIVA [58], and so on. As these datasets lack the depth modality and the corresponding 3D label, they cannot be used to evaluate 3D hand detectors. In this paper, we proposed an open-source 3D hand detection dataset CUG Hand. As far as we know, it is the first RGB-D hand detection dataset recorded in unconstrained environments. In the CUG Hand dataset, light conditions such as back-light and dark light are considered to evaluate the robustness of 3D hand detectors.

Hand pose estimation dataset contains depth images with 3D labels, so it can be used for hand detection, e.g., NYU [36], BigHand2.2M [59], ASTAR [34], EgoDexter [60], RHD [21] etc. Nevertheless, the environments of these datasets are constrained and the hands can be easily detected. The hands are normally assumed to be the nearest objects to the camera. Among the existing RGB-D hand pose estimation datasets, RHD dataset [21] is more complex than others in terms of hand detection, and some state-of-the-arts also report their hand detection results on RHD dataset. Thus, we also evaluate our method on RHD dataset to compare with the state-of-the-arts.

## 3. Methods

The paper aims at detecting 3D hand from a single RGB-D image in unconstrained environments. We list the assumptions of this study as follows: (1) We assume that the RGB-channel and the D-channel are aligned pixel-wisely in the RGB-D image. Fortunately, most consumer-level RGB-D cameras (such as Kinect, Intel realsense, and so on) satisfy this assumption. (2) We assume that the D-channel has been calibrated. As far as we know, most consumer-level RGB-D cameras have been properly calibrated in factory.

The framework of the proposed approach is shown in Figure 4. The input of the network is a RGB-D image in which the D channel is transformed into stacked sub-range representation. To extract fused RGB-D features (shared features) from the RGB branch and the SSR branch, an adaptive fusion network (AF-Net) is designed. The shared features are then fed into a 3D hand detection module to estimate 3D locations of hands. What’s more, a hand appearance reconstruction module [12] is attached for further enhancing the generalization ability of hand detection.

The flowchart of the proposed approach is shown in Figure 5. Firstly, the D-channel is transformed to SSR, and then SSR and the RGB-channel are fed into AF-Net for feature fusion and extraction. Secondly, the extracted shared features are fed into RPN for region proposal, and then 2D hand detection is conducted. Thirdly, based on the detected 2D hand location, we estimate the 3D hand location using cascaded 3D estimation. Additionally, the dotted arrow pointing to hand reconstruction means that we reconstruct hand while training, and do not reconstruct hand while evaluating. The details of the above modules will be addressed as follows.

### 3.1. Stacked Sub-Range Representation (SSR)

Normalization is an essential pre-processing step before the input data is fed into the convolutional network. If the raw depth image is directly normalized using the whole depth range, the local depth representation will be largely weakened after normalization (please refer to Figure 6b). For example, in our experimental setup, the whole depth range is from 500 to 7000 mm, while the depth value variance corresponding to local depth representation on human hand is normally within 100 mm. As a result, the value of local depth variation on hand is less than 0.016 after normalization, and it is too “flat” to be learned by the convolutional network.

To tackle this problem, the single channel raw depth image is transformed into a multi-channel representation called Stacked Sub-range Representation (SSR), as the normalized local depth representation on hand can be enhanced by reducing the span of each sub-range (see Figure 6c). Let $R=[l_{min},l_{max}]$ denotes the whole depth range, $l_{min}$ and $l_{max}$ denote the min and max values of *R* respectively. We evenly divide *R* into *k* adjacent sub-ranges, $R^{1}$, $R^{2}$, …, $R^{k}$. The spans of sub-ranges are expanded with an overlap of 400 mm between adjacent sub-ranges, so that the integrity of hands can be ensured. To normalize each sub-range, given a depth value $P_{n}$, its corresponding value in *i*-th sub-range is normalized as follow,
(1)$$P_{n}^{i}=\left\{\begin{array}{cc} \;1 & ,l_{max}^{i}\leq P_{n}^{i}\\ \frac{P_{n}-l_{min}^{i}}{l_{max}^{i}-l_{min}^{i}} & ,P_{n}\in R_{i}\\ \;0 & ,l_{min}^{i}>P_{n} \end{array}\right.,$$
where $P_{n}^{i}$ is the normalized value in the *i*-th sub-range, and $l_{min}^{i}$ and $l_{max}^{i}$ are the min and max values in the *i*-th sub-range respectively.

### 3.2. Adaptive Fusion Network (AF-Net)

The D channel is transformed into SSR. The RGB-D fused features (shared features) are learned from RGB channels and SSR channels using AF-Net. The RGB feature and depth feature are extracted from RGB and SSR branches in parallel. Then, the RGB feature and the depth feature are adaptively fused by AF-Net. The multi-scale fused features are further fused in the same manner as FPN [61].

#### 3.2.1. Adaptive Fusion Unit (AFU)

The RGB feature contains richer information and is more effective comparing to the depth feature, so it is not wise to equally fuse the RGB and depth features. As we cannot manually quantify the importance of RGB feature and depth feature, the RGB-D fusion weights are hard to be determined. Based on the above observations, we propose AFU to fuse the RGB-D features. In the fusion process, the RGB feature is the dominant and the depth feature is complementary, and these features are fused adaptively.

AFU assigns a weight to each channel of the depth feature. Given a paired RGB feature $F_{rgb}=\left\{f_{rgb}\left[c\right]|c=1,2,\ldots ,C\right\}$ and depth feature $F_{d}=\left\{f_{d}\left[c\right]|c=1,2,\ldots ,C\right\}$, where f[c] is the *c*-th channel of the feature map and *C* is the number of the channels, AFU fuses the two features with weights $\Omega =\left\{\omega_{c}|c=1,2,\ldots ,C\right\}$. The fused feature $F_{rgbd}=\left\{f_{rgbd}\left[c\right]|c=1,2,\ldots ,C\right\}$ is computed as,
(2)$$f_{rgbd}\left[c\right]=f_{rgb}\left[c\right]+tanh\left(\omega_{c}\right)\cdot f_{d}\left[c\right].$$

Specifically, we use tanh(·) function to constrain the weight of $f_{d}\left[c\right]$ to the range of [−1,1].

#### 3.2.2. Feature Extraction and Fusion

Firstly, AF-Net gradually extracts RGB and depth features using two individual deep residual network (Resnet [62]). Similar to [61], we divide Resnet into 4 stages. These stages are represented by the yellow arrows in Figure 4. The output features of these stages are denoted as $\left\{F^{1},F^{2},F^{3},F^{4}\right\}$ which are down-sampled with strides of $\left\{4,8,16,32\right\}$ and used for RGB-D fusion.

Before the input image is fed to AF-Net, both SSR channels and RGB channels are resized to $\left(1344,768\right)$ size. The depth feature $F_{d}^{i}$ generated by *i*-th stage is fed into $\left(i+1\right)$-th stage straightly. The RGB feature $F_{rgb}^{i}$ is fused with the depth feature $F_{d}^{i}$ by AFU, which generates the fused feature $F_{rgbd}^{i}$. Then $F_{rgbd}^{i}$ is fed into $\left(i+1\right)$-th stage.

Next, all the fused features $F_{rgbd}$ are processed with a conv 1×1 to unify the channels of the fused feature, and fused features $F_{f}$ are generated. Finally, the up-sampled higher fused feature $F_{f}^{i}$ is fused into lower fused feature $F_{f}^{i+1}$. After that a conv3×3 is performed, to obtain the shared feature $F_{s}^{i+1}$. All of the shared features will be taken as the input of 3D hand detection module.

### 3.3. 3D Hand Detection

The origin of the RGB-D image is on the top-left of the image, and the origin of the depth (or to say the 3D point-cloud) is the origin of the camera coordinate frame. Our goal is to construct a 3D hand detector that estimates the 3D hand location $\left(x_{1},y_{1},x_{2},y_{2},z_{c}\right)$, where $\left(x_{1},y_{1}\right)$ and $\left(x_{2},y_{2}\right)$ are the top-left and bottom-right vertices coordinates of the bounding box of a hand in the image, and $z_{c}$ is the 3D location of the center of hand along z-axis of the camera coordinate frame. We don’t estimate the thickness of a hand, because existing depth camera cannot capture the hand’s back-side facing away from the sensor.

We focus on constructing a 2D CNN based framework to extract shared features from which $\left(x_{1},y_{1},x_{2},y_{2},z_{c}\right)$ can be estimated. To simplify the task, we split it into two sub-tasks: the first is to detect 2D bounding box of hand $\left(x_{1},y_{1},x_{2},y_{2}\right)$, and the second is to estimate the 3D hand location along z-axis $\left(z_{c}\right)$ based on the 2D bounding box proposed by the first sub-task. For the first sub-task, we firstly apply region proposal network to propose a large number of 2D hand candidates. From these 2D hand candidates, we pick out those which are probably hands. Simultaneously, Offsets are estimated to compensate the 2D locations of those candidates for higher accuracy. For the second sub-task, we propose a cascaded 3D estimation network to precisely estimate $z_{c}$ the hand 3D location along the z-axis.

#### 3.3.1. Region Proposal Network (RPN)

Region proposal network (RPN) is an effective algorithm to generate 2D candidates. Comparing to selective search [63] and objectness [64] algorithms, RPN is faster with less computational consumption and reasons faster. What’s more, it can be trained with other network jointly for efficiency.

RPN applies 15 anchors to capture hands. Each anchor can be represented by $\left(x_{c},y_{c},w,h\right)$, where $\left(x_{c},y_{c}\right)$ is the center of the anchor and (w,h) are width and height of the anchor. A bounding box$\left(x_{1},y_{1},x_{2},y_{2}\right)$ can be transformed to $\left(x_{c},y_{c},w,h\right)$ as follows
(3)$$\left(x_{c},y_{c},w,h\right)=\left(\frac{x_{1}+x_{2}}{2},\frac{y_{1}+y_{2}}{2},x_{2}-x_{1},y_{2}-y_{1}\right)$$

Instead of estimating the 2D bounding boxes$\left(x_{1},y_{1},x_{2},y_{2}\right)$ of hands, RPN estimates the offset $\left(x_{c,o},y_{c,o},w_{o},h_{o}\right)$ between a anchor and ground true bounding box. The candidates outputed by RPN can be calculated as follow
(4)$$\left(x_{c}^{r},y_{c}^{r},w^{r},h^{r}\right)=\left(x_{c},y_{c},w,h\right)+\left(x_{c,o},y_{c,o},w_{o},h_{o}\right)$$

Taken the shared feature as input, RPN processes the shared feature with a conv 3×3. Then, two conv1×1 are appended with one to calculate class socre (cls) to determine whether the proposal is a hand and the other to estimate the offset between anchor and ground true bounding boxes. In training, we take a candidate as a positive sample, if the intersection over union (IoU) between a candidate bounding box and a ground true bounding box is bigger than 0.75. It is a negative sample, if IoU is smaller than 0.25. Hereof, IoU is defined as
(5)$$IoU=\frac{Box_{1}\cap Box_{2}}{Box_{1}\cup Box_{2}}.$$

At last, a non-maximum suppression (NMS) is implemented to remove those highly overlapped candidates.

#### 3.3.2. 2D Hand Detection

In this section, we present estimation of 2D locations$\left(x_{1},y_{1},x_{2},y_{2}\right)$ of all the hands in a RGB-D image. Firstly, the shared feature from AF-Net is fed to RPN to generate 2D candidates. A refine process is needed for excluding false positive candidates and improving the accuracy of 2D locations of hands.

Given a candidate $\left(x_{c}^{r},y_{c}^{r},w^{r},h^{r}\right)$ from RPN, we transform it to $\left(x_{1}^{r},y_{1}^{r},x_{2}^{r},y_{2}^{r}\right)$ using Equation (Equation 3). A bi-linear pooling with 2D candidates is used to accurately crop the shared feature for region of interesting (ROI) features with unifying sizes, because $\left(x_{1}^{r},y_{1}^{r},x_{2}^{r},y_{2}^{r}\right)$ is float, and each candidate has different aspect ratios.

After that, ROI features are fed to a fully connection module (FC) to deduct the offset $\left(x_{1}^{r,o},y_{1}^{r,o},x_{2}^{r,o},y_{2}^{r,o}\right)$ of the candidates and exclude false positive candidates. The output of the FC module are two vectors. One of them is a two-dimensional soft-max probabilities of hand and background. The other is the offset $\left(x_{1}^{r,o},y_{1}^{r,o},x_{2}^{r,o},y_{2}^{r,o}\right)$ of the candidates. Finally, we get the 2D locations of hands using 2D offsets to compensate 2D candidates as follows
(6)$$\left(x_{1},y_{1},x_{2},y_{2}\right)\;=\;\left(x_{c}^{r},y_{c}^{r},w^{r},h^{r}\right)\;+\;\left(x_{1}^{r,o},y_{1}^{r,o},x_{2}^{r,o},y_{2}^{r,o}\right).$$

#### 3.3.3. Cascaded 3D Estimation

We aim to estimate the 3D location $\left(x_{1},y_{1},x_{2},y_{2},z_{c}\right)$. From the module introduced above, we get $\left(x_{1},y_{1},x_{2},y_{2}\right)$. What we need is the 3d location along the z-axis $\left(z_{c}\right)$. We construct a cascaded 3D estimation module to estimate the corresponding $z_{c}$ to 2D locations of hands from the D channel. This module is pretty light and can be used to estimate $z_{c}$ precisely. In this module, the D channel is cropped by the estimated 2D locations$\left(x_{1},y_{1},x_{2},y_{2}\right)$ of hands. Then after resizing the cropped D channel to $\left(28,28\right)$ size, we get D patch for estimating $z_{c}$. We use a small and efficient CNN network, composed of four convolutions, to estimate the $z_{c}$.

In the first step, the CNN network estimates coarse 3D location along the z-axis $\left(z_{c}^{r}\right)$ from the patches. In the second step, each depth patch is firstly deducted by $\left(z_{c}^{r}\right)$ predicted above and then fed to the network for the offset $z_{c}^{r,o}$, to improve the depth accuracy. We get the 3D location along the z-axis $\left(z_{c}\right)$ as follows
(7)$$z_{c}=z_{c}^{r}+z_{c}^{r,o}.$$

With 3D locations$\left(x_{1},y_{1},x_{2},y_{2},z_{c}\right)$, hands in RGB-D image are located.

### 3.4. Hand Reconstruction

In order to enhance the detection accuracy and generalization ability, hand reconstruction is attached as a auxiliary task [12]. The hand reconstruction module reconstruct the hand appearance of the RGB channels and SSR channels from ROI feature.

In the module, the shared feature is fed to a conv1×1 to estimate the mean μ and logarithmic standard deviation σ. Then, the latent vector *g* is calculated with μ, σ and a standard Gaussian distributed noise Φ as following function
(8)$$g=\mu +\frac{e^{\sigma }}{2}\times \Phi .$$

Finally, we apply a deconvolutional module to generate reconstructed hand images. For more details about hand reconstruction, please refer to [12].

## 4. Cug Hand Dataset

The dataset is collected using an Intel RealSense D435i depth camera which captures RGB-D images with a resolution of 1280×720. The camera is calibrated in factory, and the calibration information can be retrieved through pyrealsense2 SDK (Software Development Kit) v2.33.1. The FOV (Field Of View) of the camera is $69^{\circ }\times 42^{\circ }\left(Horizontal\times Vertical\right)$. Specifically, for the device we use, the focal length is 919 pixels, and the principal point is (649, 355) pixels. Noted that, different devices of the same model may have different parameters, as they are calibrated individually in factory. The depth range of this camera is within 10 m. The depth accuracy is related to the distance of the object. The depth error increases as the distance increases. After the factory calibration, the depth error along the distance of this camera is less than one percent of the distance from the object. As the hand is too small to be identified when its distance to camera is further than 7 m, all hand instances are collected within 7 m. The depth image is aligned with the RGB image pixel-wisely. The RGB-D images are collected from 27 distinct subjects. The number of subjects on a single image varies from 1 to 7, and the maximum number of hands per image is 8. The distances from the hand instances to the camera range from 500 to 7000 mm, and the area of the hand bounding boxes varies from 238 to 73,062 ${\mathrm{pixel}}^{2}$.

The RGB-D images are classified into following cases according to the complexity of the scene: (1) simple case, in which only single hand appears; (2) ordinary case, with less then 4 hands; and (3) complex case, with more than 4 hands and clutter background. Examples of the above mentioned cases are shown in Section 5.2.6. Furthermore, we also consider extreme light conditions such as (4) back-light case, in which the camera faces the light source, and (5) dark case, in which there is almost no environment light. Examples of the extreme light conditions cases are shown in Section 5.2.6.

In total we collected 1244 RGB-D images, in which 625 images (3040 hand instances) are used for training, and 619 images (2334 hand instances) are used for testing. Both the training and testing sets contain ordinary and complex cases. In order to evaluate the robustness and the generalization ability of the learned model, we include unseen lighting conditions (i.e., back-light and dark cases) in the testing set only, which results in a challenging evaluation benchmark. The numbers of images in these cases are listed in Table 1.

## 5. Experiments

### 5.1. Experimental Settings

The experiments are performed on a desktop PC with an Intel i7-8700 CPU, and a Nvidia RTX 2080Ti GPU. The algorithm is implemented by python. The model is trained using a SGD Optimizer with initial learning late of 0.003, momentum 0.9 and weight decay 0.0005. An adaptive learning rate schedule is used to multiply the learning rate by a factor of 0.1 every three epochs. For SSR, the whole depth range [500, 7000] is evenly divided into three sub-ranges with 400 mm overlaps between the adjacent sub-ranges. Specifically, the sub-ranges are [500, 2867], [2467, 5034], and [4634, 7000] respectively.

### 5.2. CUG Hand Dataset

The 3D hand detecting result contains two parts: The 2D location on image plane and the 3D location along the z-axis of camera. To compare with the state-of-the-arts, we first evaluate the 2D detection accuracy on image plane, and then evaluate the 3D detection accuracy along the z-axis (please refer to Section 5.2.5). Similar to previous papers [11,12,62,65], we apply AP to evaluate the 2D detection accuracy. The IoU threshold of AP is 0.5 by default in the following experiments. We also evaluate the overall performance of the compared methods using precision recall (PR) curve which describes the relation between the precision and the recall. The precision is the ratio between the True Positive (TP) samples and all the positive samples, and the recall is the ratio between the TP samples and all the detected samples.

The following methods are compared in this paper: (1) OpenPose [66] which is an excellent human detection approach widely used by community. (2) FPN [61]. (3) FasterRCNN [67], the backbone of our approach. (4) Xu2020 [12]. (5) Raw, which denotes depth image based detection using raw depth representation. (6) SSR, which denotes depth image based detection using SSR representation. (7) Exhaustive Enumeration, which denotes our implementation of [20]. (8) Cross-stitch, which denotes our implementation of the cross-stitch unit [55]. (9) Ours w/o reconstruct, which denotes our proposed AF-Net based approach without the reconstruction module. (10) Ours, which denotes our approach with the reconstruction module. The precision recall curves of these methods are shown in Figure 7.

The methods are classified into three categories: RGB image based method, Depth image based method, and RGB-D fusion based method. OpenPose, FasterRCNN and Xu2020 are RGB image based methods. Among the RGB image based methods, Xu2020 performs the best, its AP is 5.9 point higher than that of the backbone FasterRCNN. Noted that OpenPose detects not only hands but also other human body parts, while our approach focuses on hand detection. Hand detection of OpenPose relies on human body detection. If body parts are occluded, the visible hands may be undetected by OpenPose. Raw and SSR are depth image based methods The backbone of Raw and SSR is FasterRCNN, the same as that of Ours. Comparing to Raw, the SSR representation significantly improve the AP by 6.5 points. The AP of depth image based methods are lower than that of RGB image based methods, because the depth images are noisy. The rest of methods are RGB-D fusion based methods whose AP are generally higher than that of the RGB and D based methods. By fusing the RGB-D channels, Exhaustive enumeration improves the AP by 2.3 points comparing to the backbone. Cross-stitch further improves the detection AP by three points. Ours w/o reconstruct improves the AP by 3.7 points comparing to Cross-stitch. The AP of Ours reaches 74.1, which is the highest among the compared methods.

#### 5.2.1. Robustness in the Unseen Cases

The AP of the above mentioned methods in the 5 test cases are shown in Table 2. While the AP of Ours is not always the highest in all the 5 testing cases, it is the robustest one. Ours achieves the highest AP in the overall test denoted by “All”, and its performance does not significantly drop in any of the 5 testing cases. Specially, the robustness of Ours is much higher than that of other compared methods in the unseen cases (i.e., the back-light and dark cases). In the dark case, the depth image based method SSR performs much better than the RGB image based methods, because the depth image is stable in dark environments while the RGB image is not. In the back-light case, the RGB image based method Xu2020 achieves the highest AP, whereas the AP of the depth image based methods drops significantly, because the depth image is unstable in the back-light case. While the AP of Ours is not the highest in these two unseen cases, it consistently performs well and its AP does not drop in both cases. Thus, Ours is robuster than Xu2020 and SSR. As for the seen test cases such as the simple, ordinary and complex cases, Ours achieves the highest AP. Overall, the robustness of Ours is the highest among the compared methods.

#### 5.2.2. Fusion Direction

In our approach, the D channel is transformed into SSR before it is fed into the network. There are two possible fusion directions: the first is to fuse the feature of the D branch into that of RGB branch, and the second is to fuse in the opposite direction. In the first fusion direction, the RGB branch is the main stream and the D branch is the complementary, as the features of the D branch is selectively fused into the RGB branch. In the second fusion direction, the D branch is the main stream and the RGB branch is the complementary. In our observation, the RGB channels plays a more important role than the D channel, because generally the RGB image based detectors work better than depth image based detectors. The default direction of Ours is “from D to RGB”. We trained two AF-Net with different fusion directions, and the evaluation results can be seen in Table 3. The “from D to RGB” direction achieves better AP and is robuster than the “from RGB to D” direction.

#### 5.2.3. Reconstruction Module

As can be seen in Table 4, the reconstruction module helps to improve the accuracy of the proposed detection approach. There are 4 options: The first is without reconstruction, the second is to reconstruct the hand appearance of the SSR channels, the third is to reconstruct the hand appearance of the RGB channels, and the last is to reconstruct the hand appearance of RGB-SSR channels. Reconstructing the SSR channels alone does not improve the AP. Reconstructing RGB channels slightly improves the AP. Furthermore, reconstructing the RGB-SSR channels helps improve the AP by 1.9 points, and the robustness in unseen cases is also enhanced comparing to Ours without reconstruction.

#### 5.2.4. The Number of Sub-Ranges in SSR

More sub-ranges in SSR is not always better. The AP of Ours w/o construct with respect to *k* (the number of sub-ranges in SSR) is shown in Figure 8. When k=1, SSR is equivalent to the Raw representation. When k≤3, the AP increases as the number of sub-ranges in SSR increases. When k=3, the AP reaches its peak. When k≥4, the AP drops as the number of sub-ranges in SSR increases. With the increase of *k*, the span of sub-ranges is reduced, and the normalized local features within each sub-ranges is enhanced. However, the amount of pixels within each sub-range is reduced with the increase of *k*. It turns out that k=3 is the optimal point for the SSR representation.

#### 5.2.5. 3D Hand Location Estimation on Z-Axis

In this work, 3D hand detection is conducted by firstly detecting 2D hand and then estimating 3D hand location. After the 2D bounding box of hand is located, the 3D location on z-axis is estimated. There are two options for 3D hand detection: The first option is to directly regress the hand 3D location on z-axis, and the second option is to estimate 3D location using the cascaded network. The mean errors of the two options are shown in Table 5. The overall detection accuracy of the cascaded network is significantly higher than that of the direct regression. Furthermore, the 3D detection error is closely related to the distance from hand to camera, because the depth noise in D channel increases as the distance increases. The longer the distance is, the bigger the 3D detection error is.

The 3D location estimation accuracy is also related to the size of depth patch cropped. We list the mean error of 3D location estimation with respect to the patch size in Table 6. It is observed that, when the patch size is 28×28, the mean error is the lowest. Therefore, we set the patch size as 28×28 in our experiments.

#### 5.2.6. Qualitative Results

The qualitative hand detection results are shown in Figure 9. Our proposed approach reliably detect multiple hands in unconstrained environments. In Figure 9, the green boxes denote the detection results of Ours, and the red boxes denote the ground truth labels. We observe that most of the hands are correctly detected, as the green boxes precisely cover the corresponding red boxes. Failure cases are shown in Figure 9f: A green box covers its corresponding red box, but the IoU between these two boxes is low, so that it is counted as a false detection; a red box is not covered by any green box, and it is counted as a missing detection. Figure 9f shows that, there would be false or missing detection when the hands are partially occluded by other skin color body parts.

Hand detection results of the compared methods in the dark case are shown in Figure 10. As the RGB image becomes dim in the dark case, the performance of the RGB channel based method Xu2020 degrades. We also compare the hand detection results in the back-light case, as can be seen in Figure 11. Since invalid regions may occur in the D channel, the D channel based method SSR does not work well in the back-light case. Our proposed RGB-D fusion approach effectively fused the color feature and the depth feature, which is more robust than other methods in unseen lighting conditions.

### 5.3. RHD Dataset

RHD hand dataset is proposed by [21] for hand pose estimation. Among the existing RGB-D hand datasets, RHD is more complex than the other datasets in terms of hand detection, thus sometimes this dataset is also used for hand detection evaluation. The hand detection results on this dataset are reported by some state-of-the-arts [21,68,69]. The evaluation metrics of the reported results are IoU score, Precision, Recall, and F1-score. In order to compare with the state-of-the-arts, we also evaluate our method using the same metrics as the previous methods use. The detection results are shown in Table 7. The F1-score of Ours is 95.23, significantly higher than that of other compared methods.

The performance of Ours on RHD dataset is reaching saturation, and there remains very limited space for further improvement. This observation suggests that RHD dataset is not difficult enough for RGB-D hand detection evaluation, and more challenging hand detection dataset is required. On the contrary, the F1-score of the most accurate approach on CUG Hand dataset is only 74.97 which is much lower than that of RHD dataset. It suggests that, the CUG Hand dataset is much more difficult than RHD dataset. As various challenging factors such as extreme light conditions, hand shape, scale, view point, partial occlusion, and so on are considered in CUG Hand dataset, the experimental results on CUG Hand dataset are more generic than the results on RHD dataset.

## 6. Conclusions

This paper presents a robust and accurate approach for 3D hand detection from a single RGB-D image in unconstrained environments. Empirically, our approach is evaluated on CUG Hand dataset and RHD dataset with very competitive performance. The complementary information in RGB-D channels are effectively fused by AF-Net which adaptively adjusts the fusion paths between the multi-level features extracted from the RGB-D branches. Comparing to the exhaustive enumeration fusion scheme, our approach significantly improves the detection accuracy by 8.6 points. The SSR representation improves the detection accuracy by 6.5 points comparing to the raw depth representation. We observe that the D-channel is crucial for robust hand detection. Without the D-channel, the detection accuracy of RGB-based method dramatically drops to 48.9 in unseen lightning condition, whereas our approach is robust in unseen lighting conditions. The proposed approach can be widely applied in many hand related applications, such as hand gesture recognition, hand pose estimation, human–computer/robot interactions, and so on. Based on this study, we plan to detect the 3D interaction among hands and objects in the future.

## Figures and Tables

**Figure 1 sensors-20-06360-f001:**
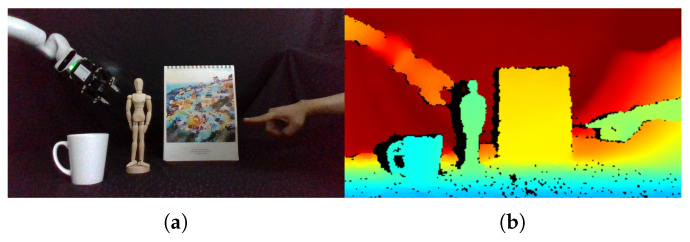
An example of 3D human–robot interaction. (**a**) The RGB image lacks 3D information, and it is difficult to distinguish which object the hand refers to. (**b**) The depth image encodes the distances from objects to the camera. It is easy to infer that the hand refers to the wooden puppet according to the depth image.

**Figure 2 sensors-20-06360-f002:**
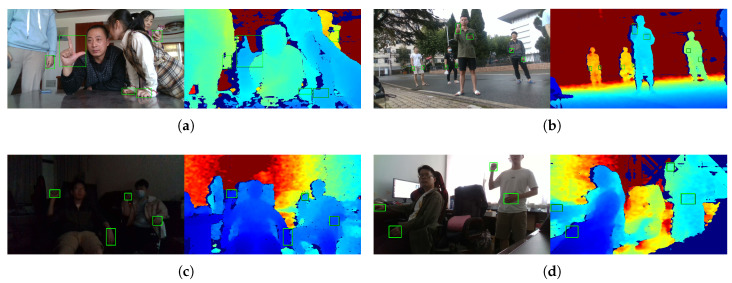
Examples of RGB-D images in our dataset. The RGB-D images are captured in various environments. For each sub-figure, on the left is the RGB channels, and on the right is the corresponding D channel. The indoor scene is shown in (**a**), and the outdoor scene is shown in (**b**). As can be seen in (**c**), the RGB channels can be significantly affected in dark conditions, while the D channel is robust. In (**d**), failure regions appear (around the upholding hand on the right) in the D channel in back-light condition.

**Figure 3 sensors-20-06360-f003:**
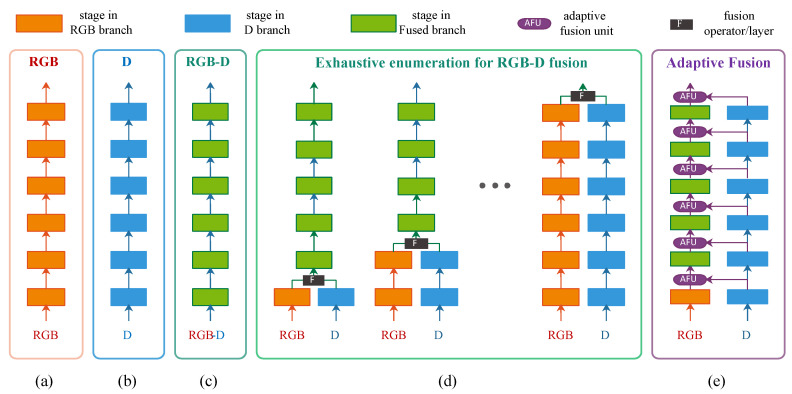
An illustration of different fusion schemes: (**a**) network for RGB channels, (**b**) network for D channel, (**c**) directly concatenation of RGB-D channels, (**d**) an exhaustive enumeration of fusion levels [20], and (**e**) our proposed adaptive fusion network (AF-Net). Noted that the figure is for illustrative purpose, and the actual number of stages in networks may vary according to different network architectures.

**Figure 4 sensors-20-06360-f004:**
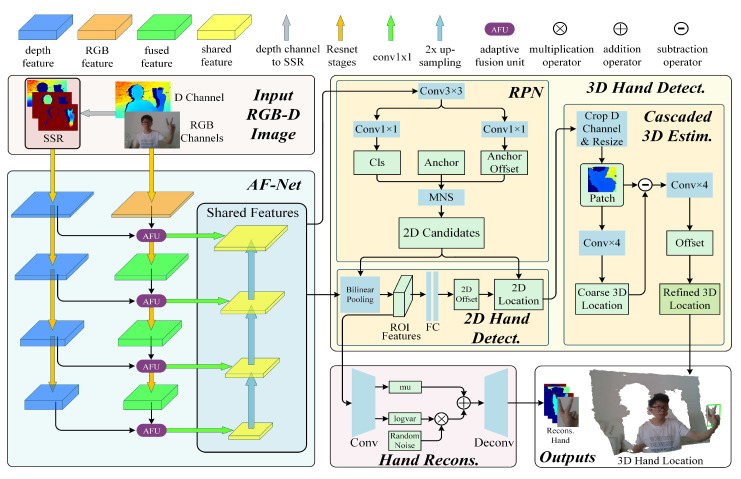
The framework of our 3D hand detection approach.

**Figure 5 sensors-20-06360-f005:**
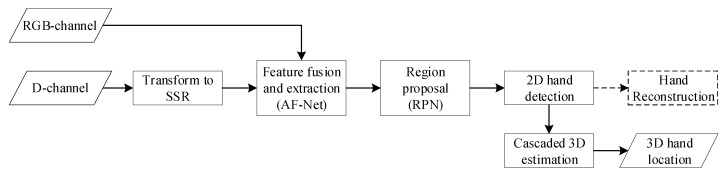
The flowchart of our 3D hand detection approach.

**Figure 6 sensors-20-06360-f006:**
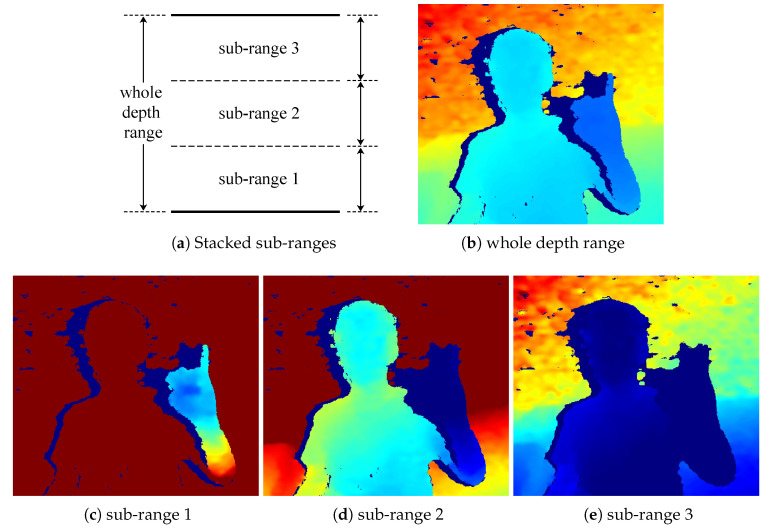
The whole depth range is evenly divided into a series of stacked sub-ranges. In (**b**), the depth image is normalized using the whole depth range. As the hand size is tiny comparing to the whole depth range, the local depth representation on hand is almost “flattened” in the normalized depth image. In (**c**), the depth image is normalized using sub-range 1. Local hand representation (detailed surface information) remains in the normalized depth image of sub-range 1.

**Figure 7 sensors-20-06360-f007:**
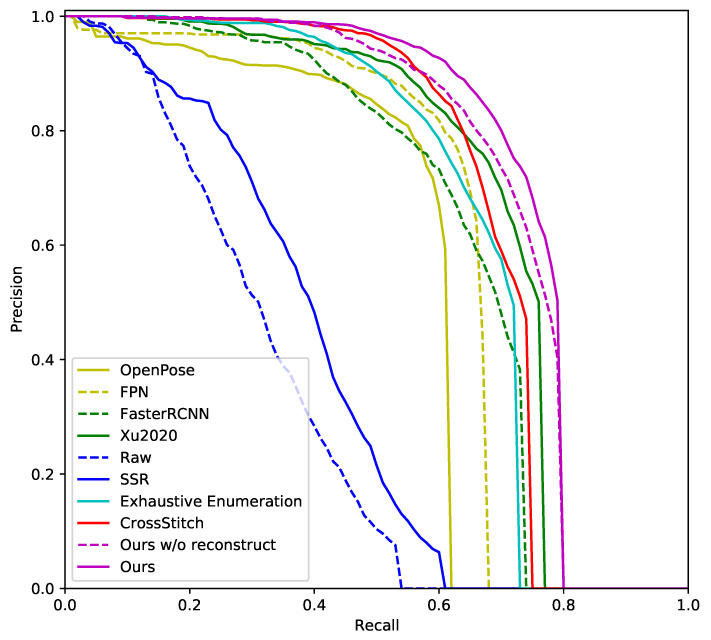
The precision recall curve of the compared methods.

**Figure 8 sensors-20-06360-f008:**
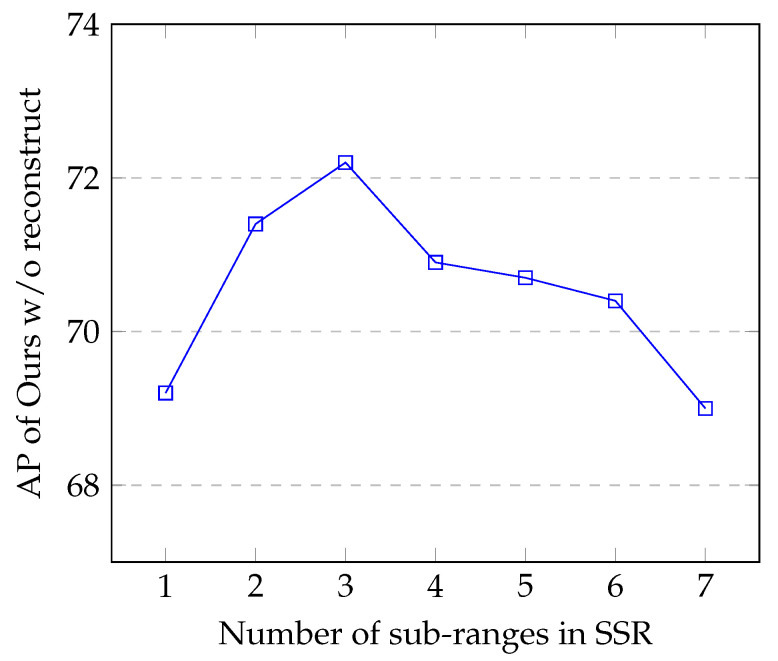
The AP of Ours w/o construct with different number of sub-ranges in SSR.

**Figure 9 sensors-20-06360-f009:**
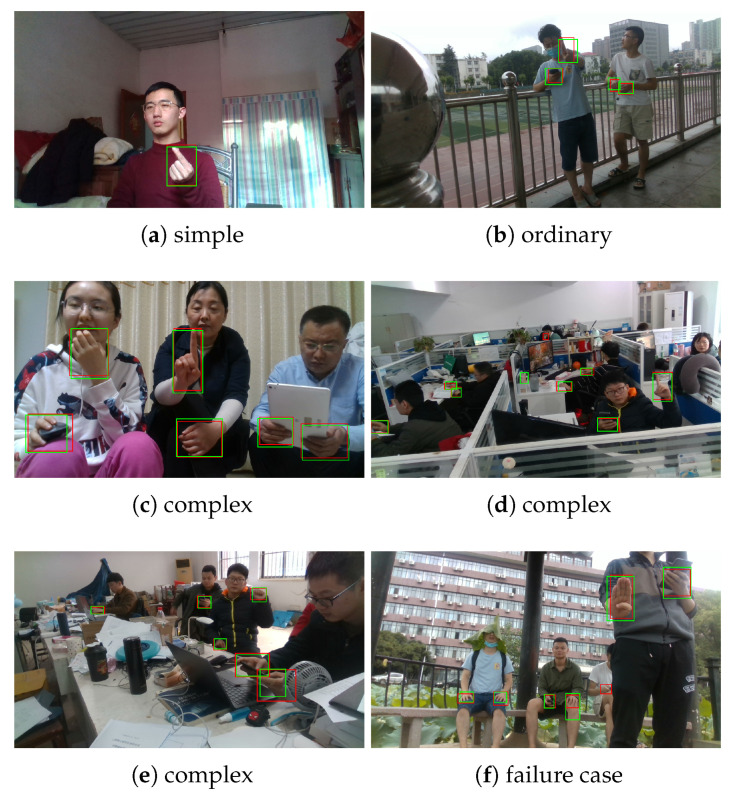
Hand detection results of Ours. The red bounding boxes denote the ground-truth labels, and the green bounding boxes denote the detection results.

**Figure 10 sensors-20-06360-f010:**
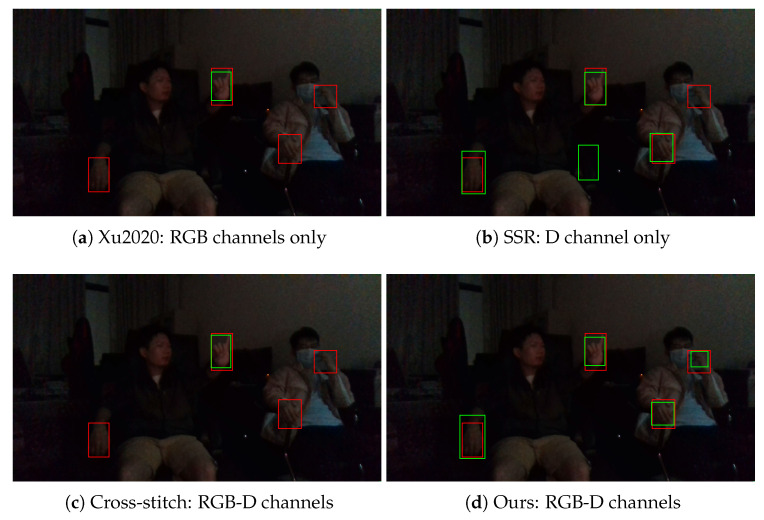
Hand detection results in the dark case. The red bounding boxes denote the ground-truth labels, and the green bounding boxes denote the detection results.

**Figure 11 sensors-20-06360-f011:**
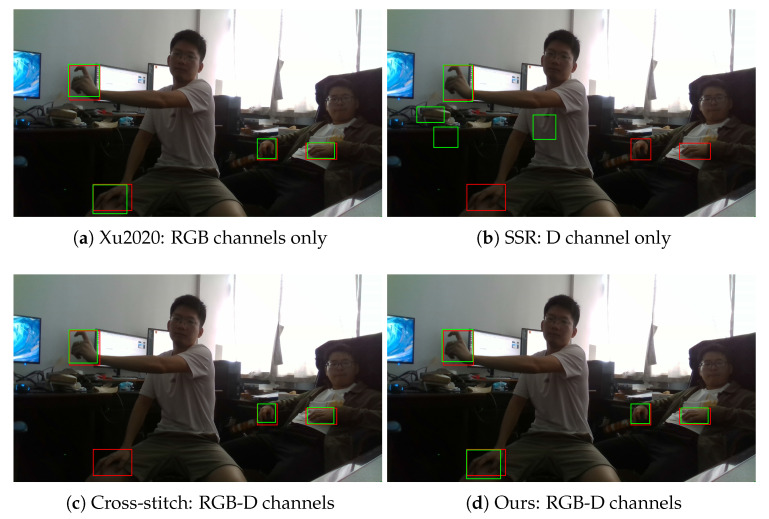
Detection results in the back-light case. The red bounding boxes denote the ground-truth labels, and the green bounding boxes denote the detection results.

**Table 1 sensors-20-06360-t001:** The number of images in CUG Hand dataset.

	Simple	Ordinary	Complex	Back-Light	Dark	All
Training	\	367	257	\	\	625
Testing	96	108	168	128	119	619

**Table 2 sensors-20-06360-t002:** Detection AP of the compared methods. The first column indicates the channels used for detection: “RGB” denotes RGB image based methods, “D” denotes depth image based methods, and “RGB-D” denotes RGB-D fusion based methods. The columns “Simple”, “Ordinary”, “Complex”, “Back-light” and “Dark” denote the five testing cases explained in the main text, and the column “All” denotes the overall test including all the cases. The red color denotes the best scores, and the blue color are the second best scores.

Channels	Method	Simple	Ordinary	Complex	Back-Light	Dark	All
RGB	OpenPose [66]	74.3	62.0	56.5	54.2	37.3	55.2
RGB	FPN [61]	100.0	78.9	63.1	56.7	27.8	61.8
RGB	FasterRCNN [67]	93.6	79.5	67.7	63.3	30.0	63.2
RGB	Xu2020 [12]	99.8	86.1	68.4	66.9	48.9	69.1
D	Raw	89.5	34.8	14.8	28.1	56.0	31.0
D	SSR	96.5	45.8	19.7	28.3	67.5	37.5
RGB-D	Exhaustive enumeration [20]	99.9	79.1	71.9	56.2	36.7	65.5
RGB-D	Cross-stitch [55]	100.0	84.0	71.3	62.5	45.1	68.5
RGB-D	Ours w/o reconstruct	100.0	87.5	71.6	65.7	58.4	72.2
RGB-D	Ours	100.0	88.0	72.7	65.9	62.5	74.1

**Table 3 sensors-20-06360-t003:** The AP of Ours w/o reconstruct with different fusion directions. The default direction is “from D to RGB”.

Fusion Direction	Simple	Ordinary	Complex	Back-Light	Dark	All
from D to RGB	100.0	87.5	71.6	65.7	58.4	72.2
from RGB to D	100.0	88.0	72.7	63.2	45.5	70.1

**Table 4 sensors-20-06360-t004:** The AP of Ours with different reconstruction options.

Options	Simple	Ordinary	Complex	Back-Light	Dark	All
w/o reconstruct	100.0	87.5	71.6	65.7	58.4	72.2
Reconstruct SSR	100.0	87.1	69.1	71.3	58.9	72.0
Reconstruct RGB	100.0	89.4	72.0	64.5	59.1	73.0
Reconstruct RGB-SSR	100.0	88.0	72.7	65.9	62.5	74.1

**Table 5 sensors-20-06360-t005:** The mean error on z-axis with respect to the distance from hand to camera. (The unit of the values in table is mm).

Distance	<1000 mm	1000–2000 mm	2000–3000 mm	>3000 mm	All
Direct regression	22.233	27.312	27.785	51.110	29.156
Cascaded network	6.396	10.619	13.998	24.253	12.154

**Table 6 sensors-20-06360-t006:** The mean error on z-axis with respect to the patch size. (The unit of the values in table is mm).

Patch Size	7×7	14×14	28×28	56×56	112×112
Direct regression	54.746	32.754	29.156	33.712	34.897
Cascaded network	26.016	16.895	12.154	13.223	14.356

**Table 7 sensors-20-06360-t007:** Hand detection results on RHD hand dataset.

Method	IoU Score	Precision	Recall	F1-Score
Christian2017 [21]	35.40	36.52	92.06	52.29
Khan2018 [68]	52.68	71.65	66.55	69.00
Baek2019 [69]	65.13	82.82	75.31	78.88
Xu2020 [12]	72.90	88.96	90.86	89.90
Ours	87.02	95.03	95.44	95.23

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
