# Peer review of "Robust 3D Hand Detection from a Single RGB-D Image in Unconstrained Environments"

_sensors, 2020, doi:10.3390/s20216360_

Round 1

Reviewer 1 Report

The content of this paper is relatively complete, some of the content needs improvement.

  1. The abstract should reflect the concrete effect of using the new method;
  2. The keywords need 4-5 words;
  3. The flowchart is needed to add for new method;
  4. The conclusion should be improved, summarizing the various research and effects for this paper.

Reviewer 2 Report

Authors could clarify the difference from the previous works. That is obviously with/without the processing of the depth information. It would be useful for readers to introduce why the depth information is critical for the hand detection (and consequently improve the performance).

Then, it is important to describe how critical the accuracy of the depth reading, thus, the calibration of the depth sensor is.

In addition, how the depth sensor deal with the “unknown” distance, e.g. infinity. Normal RGB sensors (camera) capture any light from infinity distance. If the depth sensors may have any limit, how can the two image information be fused? Authors seemed to have prior knowledge of the range of the depth sensor, but it is unclear how that was obtained and validated.

Introduction stated three open questions that the present study would address. In relation to the first point, “to determine which level(s) of features are optimal for RGB-D fusion”, the definition of “features” could be given.

The list of the contributions seemed to be very specific to what the proposed algorithm would perform. It would be good to list more generic contributions to scientific and engineering communities if applicable.

As with the Contribution 2, it is unclear what meant “D-channel is converted to SSR”.  Practically, the depth information was classified into three, rather than “converted”.

As with Contribution 3: it is unclear how the limit was originated.

Related works were given in details but in bullet points. The descriptions could be simplified to give readers overview of the relevant works. In particular, the relation to the pose detection algorithms, whether they may be applicable to the hand detection, if not, why not, could be commented. There have been studies for sign language detection. Please comment whether the present study may be applicable.

Method: It would be good to have a list of any assumptions in the present algorithm, if any.

The story of the manuscript could go along with the Figure 4, but not all details were covered.

Some more details for the logical background and definitions could be inserted. For example, depth sub-range three was how defined, But no explanation about why three, and how the overlap was introduced (in Experiment).

Eq(2): please explain why “tanh” was used.

Section 3.2.2: Please comment on how the dimension of convolution pyramid was defined and whether it may be relevant to the size restriction of the input data. Please comment on how the origin of the image and depth was defined or set, which should be relevant to define the Offset.

Section 3.3.3: Please comment on how the size (28,28) was determined.

Eq(8): Please explain why the logarithmic standard deviation was used, and how the Gaussian noise was constructed (its mean and sd). “c”: the same symbol as the channel should not be used?

4.CUG Hand Dataset: please insert more comments on what variation of the images was included; for example, angles of hands, size, colors, the same scene under different illuminants, and any hands but wearing gloves (non natural skin color).

5. Experiment

5.2, Fig 6: please clarify what precision was measured across the different algorithm. It is unclear what detection was made by each and whether those are comparable: e.g. Raw is only for depth.

Please clarify the unit of AP. Usually AP ranged 0-1 but authors commented as “its AP is 5.9 point higher than…”.

Table 2: if the difference between the present study and author’s previous (Xu2020) are only the use of the depth information,  if the present method allied to the RGB data (channels), how different the results are?

Please comment whether the present algorithm is tested with applying to the RGB dataset (ignoring the depth), as a control.

Please also comment whether any image with false hand, such as hand-shaped leaves, and how the performance was with close-up image of hand (would this be included in Simple?).

Table 3 and 4: please clarify whether any normalisation was undertaken to explain the AP for Simple are all 1.00.

Table5: Please clarify what results were shown in the table. Text says detection performance, but the table has “mm” distance.

5.2.6: please clarify what judgements or tasks were taken for the “qualitative” results.

Table6: please clarify how the performance depends on the database, or more generic in comparison to the results in Tables 2-5.

Reviewer 3 Report

This paper proposed a 3D hand detection approach which improves the robustness and accuracy by adaptively fusing the complementary features extracted from the RGB-D channels. And the novel method and analysis results of the paper are described in detail. The specific opinions are as follows:

1.The related work summarizes the current research progress of energy supply, but the following references (with certain topics) should be added to the content and discussed in where applicable.

Semantic relation extraction using sequential and tree-structured LSTM with attention.

2. In the comparison part of the results, it is better to add other evaluations algorithms and the algorithm designed in this paper for comparison experiment.

3.The analysis result about more datasets should be added in detail.

4.English impression has to be improved to avoid misunderstandings by readers.

Round 2

Reviewer 2 Report

There are some minor points which authors could address or edit.

Section 1. Authors could comment how important to detect hands in static images (rather than motion or video).

Section 3. Authors mentioned that the depth sensor (camera) was calibrated but authors could comment more, that is, exactly what were calibrated: e.g. precision of depth along with distance and/or viewing angle.

Section 4. Authors could inform more about the equipment and edit the information:i.e. insert the origin of country of the depth camera; insert the unit of the spatial resolution "pixel"; what is the range of the depth, field of view, and  and error range of the depth, distance, and field of view (these could be relevant to the information of the calibration in Section 3).

The "pyrealsense2 sdk" should be explained, as this is part of "equipment": i.e. manufacture, version/model. the origin of country.

This may be obvious, but does "sdk" sand for "software development kit"?

Is this based on Python (C++, or Matlab)?
